# Who behaves more pro-environmental in the national parks: A comparison of the tourist and the hiker

Qing Zhang[1]*, Huazhen Sun[1,2], Xiasui Peng[1], Qiuyan Lin[1]

**1** School of Tourism, Wuyi University, Nanping, China, **2** Graduate School of Business, SEGi University, Kuala Lumpur, Malaysia

* jamking5@163.com

## Abstract

The intention of pro-environmental behavior (PEB) directly affects the sustainable development of protected areas, especially national parks, but few studies have done comparative research on tourist and hiker behaviors. This study explores the intention of tourists' and hikers' pro-environmental behavior based on theory of planned behavior (TPB) and norm activation theory (NAM). Researchers surveyed 454 tourists and 466 hikers in Wuyishan National Park a structural equation modeling data analysis method. The results demonstrate that the TPB and the NAM were accurate in describing for tourists' and hikers' pro-environmental behavior in national park. However, for specific influencing factors, hikers' attitude, awareness of consequences, and assumption of responsibility were significantly different from those of the tourists. This study sheds light on how to better comprehend and advocate for PEB in national parks and proposes different management approaches to improve the PEB of tourists and hikers.

**Data Availability Statement:** The data of this study share on the figshare web with the DOI:10.6084/m9.figshare.23295461.

**Funding:** The authors received no specific funding for this work.

## Introduction

From the standpoint of sustainability, tourism PEB has spread gained increasing popularity globally [1–5]. Previous studies have shown that irresponsible environmental behavior leads to adverse impacts on the environment [6, 7]. Importantly, environmental quality impacts tourist destinations' attractiveness to visitors, as well as tourist satisfaction [8, 9]. China is a vast, emerging tourism destination in which more than 67% of land area is comprised of mountainous regions [10]. The mountain is particularly precarious with more fragile, ecological environment, and it is more difficult to collect and transfer litter [2, 3, 11]. As of July 2022, China had a total of 5 National Parks: Sanjiangyuan, Panda, Northeast Tiger and Leopard, Hainan Tropical Rainforest, and Wuyishan [12]. The national parks need to be profitable and environmentally friendly. Hiking is one of the most popular and interesting outdoor hobbies [13–15], and one of the most basic and important activities in national parks and other outdoor recreation areas. Although hiking was once considered a niche activity, it has become a popular outdoor sport [16]. It is believed that walking in mountainous areas with different altitudes, can bring many benefits to the participants [17]. However, China now has 60 million hikers, which

**Competing interests:** The authors have declared that no competing interests exist.

has a number of negative effects on the ecosystem [18–20]. This has resulted in worsening conditions and environmental challenges, as picturesque mountain areas draw significant numbers of hikers [21, 22]. Therefore, enhancing the PEB of Chinese hikers has become an important issue that must be addressed urgently [7, 22, 23].

Scholars applied different theories to study PEB, including the theory of planned behavior (TPB) [24–33], attitude behavior context theory [34–37], goal-directed behavior theory [38–44], norm activation model (NAM) [27, 43, 45–51], and value belief norm theory [9, 37, 52–57]. TPB and NAM have been most widely used to study PEB, and are considered to be reliability models [7, 30]. The existing literature on PEB can be categorized into two primary perspectives; rationality- and morality-based approaches [43, 58]. Some researchers opt for rational-choice models, analyzing PEB with TPB [59, 60], while others argue that PEB is primarily influenced by morality and adopt NAM [61–63]. Notably, very little research has addressed comparative PEB in different groups of people.

Furthermore, there have been few studies to examine hikers' PEB, particularly in popular locations like China [64, 65]. Traditional visitors have been used as research subjects in the majority of previous studies, but hikers' present unique characteristics; for example, hikers are typically better educated and more environmentally conscious [66], supporting nature conservation and valuing natural beauty and landscapes [67, 68]. Nonetheless, hikers' PEB, particularly the differences between the tourist and hikers, are yet to be explored. It is unclear which factors are most important in influencing the PEB between tourists and hikers. In addition, research has not paid significant attention to hikers' PEB, particularly in protected regions, like national parks, and there are few articles that take Chinese hikers as research objects, as most of the research investigating PEB in protected areas has been done within Western cultures, such as the USA, Australia and Canada [5].

Here, TPB and NAM were utilized to examine the influencing factors of tourists' and hikers' PEB in order to close these research gaps. This study used Wuyishan national park as a case study, taking token littering, stepping on grass, picking of flowers, and breaking of branches, plants and trees as irresponsible environmental behavior [22]. The main questions of this study were the following: (1) What are the main factors that impact tourists' and hikers' PEB? (2) Which model is a better fit for tourists, and which model is a better fit for hikers? (3) What are the relationships between these factors? The study attempts to develop an effective theoretical framework to explain the factors affecting hikers' PEB and propose useful management suggestions to achieve environmentally friendly development in protected areas.

Based on these gaps in the published research, TPB and NAM were applied to study factors influencing tourists' and hikers' PEB in the mountains in a Chinese national park, focusing on seven variables (attitude, subjective norm, perceived behavioral control, awareness of consequences, personal norm, ascription of responsibility, behavioral intention). This study made an in-depth comparison of the PEB of tourists and hikers, using TPB and NAM for detection, and comparative analysis of the factors in the models. Finally, we propose different management approaches for tourists and hikers to improve PEB.

## Literature review and hypotheses development

### Research hypotheses

**(1) Attitude.** Attitude (ATT) toward behaviors refer to a person's positive or negative assessments of the behaviors that are being discussed [60, 69]. A person's attitude regarding activities serves as a comprehensive psychological evaluation of their goodness, damage, pleasantness, likability, and despicability [60, 70]. Beliefs about behavior have an impact on attitude toward certain acts. Behavioral beliefs are a person's subjective assessments of his or her

actions in the world. According to the TPB, attitudes toward pro-environmental behaviors are reliable predictors of intentions to engage in ethical behavior. Correlation-based analyses have previously shown that people's attitude toward environmental behavior can predict their behavioral intention (BI) [27, 71–73]. Therefore, this hypothesis is proposed (Fig 1):

Hypothesis 1 (H1). Attitude (ATT) has a positive influence on behavioral intention (BI)

**(2) Subjective norm.** The social pressures that a person feels for engaging in or refraining from engaging in particular environmental behaviors are referred to as subjective norm (SN) [60, 69]. Fishbein and Ajzen formally added descriptive norm to injunctive norm as a second component of subjective norm [74]. Subjective norms include both injunctive norms and descriptive norms; descriptive norms are based on ideas regarding the behavior of key referents, whereas injunctive norms are based on people's perceptions of what they believe important referents (such as parents, teachers, and close friends) should do [75]. People are most likely to follow the advice or judgments of their significant others when it comes to engaging in a particular environmental activity or not, and will act in particular ways in response to the social pressure they feel. On the basis of TPB analytical framework, SN is valid predictive variables that affect BI. Correlational studies have confirmed that tourists' SN are able to predict their BI [27, 71–73]. Therefore, this hypothesis is proposed (Fig 1):

Hypothesis 2 (H2). Subjective norm (SN) has a positive influence on behavioral intention (BI)

**(3) Perceived behavioral control.** An individual's sense of how difficult or easy it is to carry out a particular environmental behavior is referred to as perceived behavioral control (PBC). It is believed that perceived behavioral control can reflect past events, upcoming challenges, and barriers [60, 69]. PBC is a result of control beliefs, which are perceptions about the presence of factors that facilitate or impede the adoption of a given behavior [75]. Perceived strength and control beliefs have an impact on perceived behavioral control, and control beliefs describe elements that might influence or prevent someone from carrying out a specific conduct. For example, a person's perception of their own strength determines whether they believe they have the power to influence external forces that could encourage or inhibit them from acting in a way that is appropriate for the situation [60, 70]. PBC legitimate predictors of BI can be based on the TPB analytical framework, and some previous literature has displayed

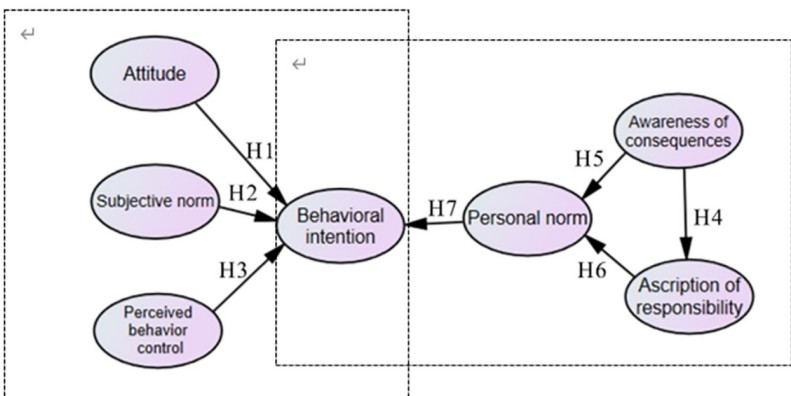

**Fig 1. Conceptual model.**

that PBC is able to predict their PEB [27, 71–73]. On this premise, the following hypotheses are proposed (Fig 1):

Hypothesis 3 (H3). Perceived behavior control (PBC) has a positive influence on behavioral intention (BI)

**(4) Awareness of consequences.** Awareness of consequences (AC) refers to "whether someone is aware of the negative consequences for others or for other things one values when not acting pro-socially" [76]. AC is alternatively dubbed problem awareness. Typically, people feel personal moral obligation to do an activity when they are aware of the detrimental effects of doing so for others. This, in turn, has a direct impact on people's intentions [77]. Within the norm activation framework, individuals' AC builds AR and PN. Correlational studies have confirmed that AC is able to predict AR and PN [5, 30, 46, 65, 78, 79]. On this premise, the following hypotheses are proposed (Fig 1):

Hypothesis 4 (H4). Awareness of consequences (AC) has a positive influence on personal norm (PN)

Hypothesis 5 (H5). Awareness of consequences (AC) has a positive influence on ascription of responsibility (AR)

**(5) Ascription of responsibility.** AR refers to the "feeling of responsibility for the negative consequences of not acting pro-socially" [76]. Accordingly, AR indicates individuals' feeling of responsibility for the consequences of pro-social acts [76]. PNs are significantly activated by both problem awareness and assigned responsibility. While PNs are directly impacted by attributed responsibility, this norm is a substantial indirect effect of problem awareness [80]. In the norm activation model, individuals' AR impacts PN. Some researchers have verified that an individual's AR influences their PN [5, 30, 46, 65, 78, 79]. Base on this, hypothesis is proposed (Fig 1):

Hypothesis 6 (H6). Ascription of responsibility (AR) has a positive influence on personal norm (PN)

**(6) Personal norm.** Personal norms are the "moral obligation to perform or refrain from a specific action" [61]. In addition, personal norms represent a moral obligation for performing a specific action or refraining from it [81]. One's pro-social intentions and activities develop as a result of problem awareness, assigned blame, and personal norms in that order of significance; personal norms are alternatively utilized with moral norms [43]. In the norm activation model, individuals' PN impacts BI. Some studies have confirmed that individuals' PNs influence their BI [5, 30, 46, 65, 78, 79]. As mentioned above, this hypothesis is proposed (Fig 1):

Hypothesis 7 (H7). Personal norm (PN) has a positive influence on behavioral intention (BI)

## Data, materials, and methods

### Study sites

Wuyishan National Park is one of the four world natural and cultural heritage sites recognized by the United Nations Educational, Scientific and Cultural Organization (UNESCO) in China. It is also in the first group of national parks in China, as it spans Jiangxi and Fujian provinces

at a longitude of 117˚24′13″-117˚59′19 and a latitude of 27˚31′20″-27˚55′49″, with the protected area covering 1001.41 square kilometers. In 2022, more than one million tourists came to visit Wuyishan National Park.

Wuyishan National Park contains a 1,160-kilometer national forest trail that starts in Liangye Mountain in Fujian Province, passes through Shangrao in Jiangxi Province, passes through Xianxia Ancient Road and Nianbadu Ancient Town at the junction of Fujian, Jiangxi and Zhejiang provinces, and extends north to the Jiulong Mountain in Zhejiang Province. The Wuyi Mountains are steep and majestic, with typical Danxia landforms; many peaks exceed 1,000 meters above sea level, and the zonal vegetation is mid-subtropical evergreen broad-leaved forest. Dahongpao is the champion of Wuyi rock tea. The Dahongpao tree in Wuyi Mountain is more than a thousand years old and is extremely rare, and Wuyi Mountain is a famous Neo-Confucian Mountain and gathering place of Hakka culture.

## Survey design

All variables were measured using scales that had been previously validated and used in environmental research. To make things simpler for the respondents to understand, some were slightly modified in accordance with the research setting. By using the back-translation method, the English scales were converted into Chinese versions and then translated back into English with a team that consisted of three professors (two Chinese and one English) to ensure content validity. A 5-point Likert scale with options ranging from "Strongly disagree" (1) to "Strongly agree" was used in the survey (5).

The questionnaire comprised two parts: the first included demographic characteristics of residents (gender, age, education, and monthly incomes); and the second was the scales for seven variables. The measurement items in present study were mainly from previous environmental behavior studies, measures of behavior attitude, subjective norm and perceived behavior control, followed by ATT1-ATT4 to measure attitude, SN1-SN3 measure subjective norm, and PBC1-PBC3 measure perceived behavior control, and BI1-BI3 measure PEB intention [21, 27, 69]. AC1-AC3measurement awareness of consequences, AR1-AR3 Attribution of measurement ascription of responsibility, PN1-PN3 measurement personal norm [61, 82–84].

## Data collection

The study was approved by the Laboratory Ethics Committee of the School of Tourism, Wuyi University (LY2022014, 4 May 2022). A pilot survey was carried out prior to the questionnaire's official publication to further assess its validity and scientific consistency. Fifty questionnaires were issued, 45 questionnaires were collected, and 5 unqualified questionnaires were excluded. The effective response rate was 80%. Based on some input on the questionnaire's design and content, it was altered to make it easier for respondents to fill it out and better grasp the information they were providing. After ambiguous items were changed, the survey's reliability was evaluated using the item-total statistics and Cronbach's alpha coefficient (alpha > 0.70). The results showed that the reliability was satisfactory.

The questionnaire was distributed to Chinese tourists and Chinese hikers by way of convenience sampling in the Wuyishan National Park: for tourists, the questionnaire was distributed in the tour area, and for the hikers the questionnaire was distributed at the trail rest point. A team of six research assistants split into two groups to gather questionnaires in-person for the offline survey from July 11 to August 14, 2022. The team was trained on how to choose respondents, employ anonymity precautions, and comprehend the study's objective. If the survey results in the sample are extended to a population of more than one million, then the study needs 384 samples [85]. Therefore, a total of 500 questionnaires for tourists were distributed

and 480 were collected, a total of 500 questionnaires for hikers were distributed and 484 were collected. After screening and excluding some invalid questionnaires with inconsistent answers, incompleteness, and the same score, there were 454 valid questionnaires for tourists and 466 valid questionnaires for hikers, with the effective response rate of the questionnaire being 90.8% for tourists and 93.2% for hikers.

## Measurement model testing

In this study, structural equation model (SEM) was used for analysis. Confirmatory Factor Analysis (CFA) was used as part of SEM analysis. Before structural equation model analysis, CFA analysis measurement model should be applied. The reduction of the measurement model variables in this study is based on the two-stage model [86]. Before doing the SEM analysis, the measurement model is tested. The second stage was carried out if it was determined that the measurement model fit was satisfactory. Completely evaluate the SEM model. Through the application of SPSS24.0 and AMOS21.0 software for analysis, the Cronbach's α coefficient of the total measurement scale in this study is 0.907 and the Cronbach's α value of each latent variable is between 0.820 and 0.896, which indicates that the scale has better reliability and internal consistency.

## Results

### Descriptive statistical analysis of the sample

We first performed descriptive statistics on the data about tourists and hikers before studying the original data obtained by the questionnaire. In the Table 1, for the tourist, the results of 454 sample surveys showed that the gender distribution of respondents was 54.8% males, 45.2% females, and slightly more males; age is concentrated between 25–54 years old (52%); monthly income is mainly 3000–7000 RMB (58.8%); most of the tourists have high school to undergraduate degree (82.5%). For the hiker, the results of 466 sample surveys showed that the gender distribution of respondents was 60.7% males, 39.3% females, and slightly more

**Table 1. Demographic data.**

| Variable | Category | N(tourist) | Percent(%) | N(hiker) | Percent(%) |
|---|---|---|---|---|---|
| Gender | Male | 249 | 54.8 | 283 | 60.7 |
| | Female | 205 | 45.2 | 183 | 39.3 |
| Age | 18–24 | 181 | 39.9 | 184 | 39.5 |
| | 25–34 | 103 | 22.7 | 104 | 22.3 |
| | 35–44 | 54 | 11.9 | 60 | 12.9 |
| | 45–54 | 79 | 17.4 | 80 | 17.2 |
| | ≥55 | 37 | 8.1 | 38 | 8.2 |
| Education level | Less than high schoo | 55 | 12.1 | 56 | 12.0 |
| | High school/Technical school | 111 | 24.4 | 115 | 24.7 |
| | Undergraduate degree | 264 | 58.1 | 271 | 58.2 |
| | Postgraduate degree | 24 | 5.3 | 24 | 5.2 |
| Monthly Income (RMB) | Under 3000 | 87 | 19.2 | 55 | 11.8 |
| | 3001–5000 | 159 | 35.0 | 150 | 32.2 |
| | 5001–7000 | 108 | 23.8 | 112 | 24.0 |
| | 7001–9000 | 76 | 16.7 | 105 | 22.5 |
| | Over 9001 | 24 | 5.3 | 44 | 9.4 |

males; age is concentrated between 25–54 years old (52.4%); monthly income is mainly 3000–9000 RMB (78.7%); most of the hikers have undergraduate degree or above (63.4%).

## Structural model testing

The results of the overall fit index of the measurement model show that χ2/df all less than 3, RMSEA all less than 0.08, GFI, AGFI, TLI, NFI, IFI, CFI (Table 2), all reaching the criterion is greater than 0.9, indicating that the overall model fits well [87]. The standard loading value corresponding to each latent variable is 0.721~0.873 and above the limitation of 0.6, all composite reliability (CR) is between 0.795~0.890 and greater than 0.6, average variance extracted (AVE) is between 0.564~0.725 and the upper limit of 0.5, which shows that the convergence validity is good [88, 89], the results showed that the reliability and the convergent validity was sufficient (Table 3). Path analysis (Table 4) shows that there is a significant influence among the latent variables. The discriminative validity table (Tables 4 and 5) also shows that the AVE root sign values of latent variables are all greater than the correlation coefficients between the various latent variables, and the discriminant degree is valid and conforms to the reference standard of [88]. The acquired data are appropriate for the measurement model, and its reliability, convergence validity, and discriminative validity are all generally acceptable.

## Correlation analysis

The maximum likelihood method is used to estimate the parameters of the structural model, and the model fitting indexes all meet the criteria. Path analysis with significance $p<0.05$ as the standard to obtain hypothesis test results (Table 3): ATT, PBC, SN have a positive effect on BI (tourist:β = 0.197, $p<0.001$; β = 0.243, $p<0.001$; β = 0.207, $p<0.05$; hiker: β = 0.422, $p<0.001$; β = 0.171, $p<0.001$; β = 0.294, $p<0.05$), H1, H2, and H3 were all supported; AR and AC have a positive impact on PN (tourist:β = 0.195, $p<0.001$; β = 0.271, $p<0.001$; hiker:β = 0.558, $p<0.001$; β = 0.318, $p<0.001$), H4 and H5 were all supported; AC has a positive influence on AR (tourist:β = 0.315, $p<0.001$; hiker:β = 0.667, $p<0.001$), H6 was supported; PN has a positive influence on BI (tourist:β = 0.443, $p<0.001$; hiker:β = 0.421, $p<0.001$), H7 was supported.

**Table 2. Goodness of fit indices of measurement model.**

| Goodness of Fit Indices | Hiker | | Tourist | |
|---|---|---|---|---|
| | TPB | NAM | TPB | NAM |
| χ2/df | 1.297 | 2.964 | 1.038 | 1.307 |
| RMSEA | 0.026 | 0.066 | 0.009 | 0.026 |
| GFI | 0.975 | 0.952 | 0.980 | 0.977 |
| AGFI | 0.962 | 0.926 | 0.969 | 0.964 |
| TLI | 0.993 | 0.951 | 0.999 | 0.993 |
| NFI | 0.976 | 0.946 | 0.983 | 0.979 |
| IFI | 0.994 | 0.963 | 0.999 | 0.995 |
| CFI | 0.994 | 0.963 | 0.999 | 0.995 |
| AIC | 140.541 | 204.217 | 125.223 | 121.329 |
| BIC | 272.32 | 319.523 | 257.837 | 237.366 |
| ECVI | 0.310 | 0.451 | 0.269 | 0.261 |

**Table 3. Results of the confirmative factor analysis.**

| Constructs and scale items | Factor loading (tourist) | CR | AVE | Factor loading (hiker) | CR | AVE |
|---|---|---|---|---|---|---|
| I think doing PEB in national parks is wise (ATT1) | 0.841 | 0.890 | 0.669 | 0.769 | 0.860 | 0.606 |
| I think doing PEB in national parks is good (ATT2) | 0.821 | | | 0.775 | | |
| I think doing PEB in national parks is worthwhile (ATT3) | 0.804 | | | 0.765 | | |
| I think doing PEB in national parks is beneficial (ATT4) | 0.806 | | | 0.804 | | |
| My friend's support my PEB (SN1) | 0.853 | 0.866 | 0.683 | 0.805 | 0.830 | 0.620 |
| People who are important to me think I should do PEB (SN2) | 0.820 | | | 0.818 | | |
| People who are important to me would want me to do PEB (SN3) | 0.806 | | | 0.737 | | |
| I have enough physical strength to participate in PEB (PBC1) | 0.845 | 0.887 | 0.724 | 0.823 | 0.879 | 0.707 |
| I am confident that I can do something helpful to protect the environment (PBC2) | 0.835 | | | 0.863 | | |
| It is easy for me to take actions to protect the environment in this national park (PBC3) | 0.872 | | | 0.836 | | |
| People's activities have negative impacts on the natural environment (AC1) | 0.855 | 0.888 | 0.725 | 0.795 | 0.827 | 0.614 |
| People's activities have negative impacts on wild animals and plants (AC2) | 0.873 | | | 0.774 | | |
| People's activities lead to water pollution (AC3) | 0.826 | | | 0.781 | | |
| All people are jointly responsible for environmental deteriorations in this national park (AR1) | 0.824 | 0.865 | 0.681 | 0.787 | 0.814 | 0.593 |
| All people are partly responsible for environmental problems in this national park (AR2) | 0.841 | | | 0.753 | | |
| All people must take responsibility for environmental problems in this national park (AR3) | 0.810 | | | 0.770 | | |
| I feel guilty for not doing PEB (PN1) | 0.829 | 0.866 | 0.683 | 0.829 | 0.832 | 0.623 |
| I think PEB is a moral obligation (PN2) | 0.820 | | | 0.752 | | |
| Pro-environmental behavior is part of my ethics (PN3) | 0.830 | | | 0.785 | | |
| I am willing to participate in PEB currently (BI1) | 0.816 | 0.842 | 0.640 | 0.721 | 0.795 | 0.564 |
| I plan to participate in PEB currently (BI2) | 0.793 | | | 0.749 | | |
| I am willing to ask my relatives and friends to participate in PEB currently (BI3) | 0.791 | | | 0.781 | | |

## Multi-group comparison

Multi-group analyses utilizing z-score comparison were carried out in order to investigate the potential moderating impact of tourists and hikers. Z-score was calculated as z = mean difference/standard error, which required the mean difference between the two distributions and standard error. When $|z| > 1.645$ and p-value $< 0.05$, there was considered to be a significant difference between the two groups [90].

**Table 4. Discriminant validity of TPB and NAM constructs for tourist.**

| TPB constructs | 1 | 2 | 3 | 4 | NAM constructs | 1 | 2 | 3 | 4 |
|---|---|---|---|---|---|---|---|---|---|
| PBC | 0.851 | | | | AC | 0.851 | | | |
| SN | 0.330*** | 0.826 | | | AR | 0.314*** | 0.825 | | |
| ATT | 0.352*** | 0.425*** | 0.818 | | PN | 0.334*** | 0.282*** | 0.826 | |
| BI | 0.375*** | 0.374*** | 0.373*** | 0.800 | BI | 0.160*** | 0.135*** | 0.478*** | 0.800 |

Note:

***p < 0.001;

The numbers in the diagonal row (bold) are the average variance extracted by each latent construct. The numbers above diagonal are the squared correlation coefficients between the constructs.

**Table 5. Discriminant validity of TPB and NAM constructs for hiker.**

| TPB constructs | 1 | 2 | 3 | 4 | NAM constructs | 1 | 2 | 3 | 4 |
|---|---|---|---|---|---|---|---|---|---|
| PBC | 0.841 | | | | AC | 0.784 | | | |
| SN | 0.387*** | 0.787 | | | AR | 0.747*** | 0.770 | | |
| ATT | 0.374*** | 0.692*** | 0.777 | | PN | 0.669*** | 0.713*** | 0.789 | |
| BI | 0.463*** | 0.694*** | 0.734*** | 0.751 | BI | 0.332*** | 0.354*** | 0.496*** | 0.751 |

Note:

***p < 0.001;

The numbers in the diagonal row (bold) are the average variance extracted by each latent construct. The numbers above diagonal are the squared correlation coefficients between the constructs.

**Table 6. Significant results of comparisons of the path coefficients between tourist and hiker groups.**

| Path | | | Tourist | | Hiker | | |
|---|---|---|---|---|---|---|---|
| | | | Estimate | P-value | Estimate | P-value | z-Score |
| BI | <--- | ATT | 0.197 | *** | 0.422 | *** | 2.787*** |
| BI | <--- | SN | 0.207 | *** | 0.294 | *** | 1.090 |
| BI | <--- | PBC | 0.243 | *** | 0.171 | *** | -1.026 |
| AR | <--- | AC | 0.315 | *** | 0.667 | *** | 4.727*** |
| PN | <--- | AR | 0.195 | *** | 0.558 | *** | 3.362*** |
| PN | <--- | AC | 0.271 | *** | 0.318 | *** | 0.486 |
| BI | <--- | PN | 0.443 | *** | 0.421 | *** | -0.332 |

Note: This table reports the unstandardized values for each group which were used to calculate the z-scores [91].

***p < .001.

The comparison's findings, shown in Table 6, indicated that the path coefficients from ATT to BI, AC to AR, and AR to PN were significantly different between the tourist and hiker groups, and the influence was stronger among hikers.

## Discussion and conclusions

### Discussion

NAM and TPB are both established theories for examining people's PEB in the context of tourism, which encompasses locations with a focus on nature. This study demonstrates that, in TPB, compare to the tourist, hikers' attitude pay more important. Attitude is the most important determinant of behavior [52]. As having a pro-environmental mindset is positively correlated with one's sense of connection to nature, hikers frequently want to act in a pro-environmental ways while they are out hiking [92, 93]. Hikers are also more likely to actively engage in environmentally sustainable buying habits because of their deeper connections to nature [94]. Customers do behave more sustainably when they have a connection to the natural world [27, 95, 96]. Compared to tourists, hikers had more perceived connections to nature, so their pro-environmental attitude was stronger. This finding differs from other literature that maintains that attitude is an insignificant predictor of behavioral intention [97].

Hikers' AC had a more significant effect AR than tourists' AC. Tourists are not familiar with the potential effects of reckless environmental activity on nature because they lack connection to nature. General attitudes toward protecting the environment can be seen as being

influenced by people's awareness of environmental implications, which is important in shaping attitudes toward pro-environmental behavior [83, 98, 99]. One of the primary components of people's attitudes toward an action is their beliefs about the effects of engaging in the conduct [100]. The PEB of people can thus be considered as being performed in a context of either usual or unusual environments [21]. Individuals' willingness to adopt PEB is contingent on the perceived consequences of such behaviors; specifically, the results of such behaviors performed in one's "usual environment" are relatively easy to be linked to one's personal interests [101]. Compared to tourists, hikers' "usual environment" includes the nature in the national park; they adopt PEB on AC. This is interpretation differs from other research.

Hikers' AR had more significant affects than tourists' PN. AR and AC come from internal desire, and has been considered to be better for predicting behavior than external stimulus [102, 103]. From the viewpoint of hikers, it can be said that when hikers take environmental issues seriously, they become more observant of how other people view or assess environmental challenges [98, 99]. "With great power comes great responsibility," hikers are more physically fit than standard tourists, therefore, hikers have more responsibility for their behavior. These findings were in accordance with the previous tourism literature indicating that internal demand was particularly important for behavior [104–106].

## Theoretical and practical implications

This study is one of the first attempts to compare the tourists' PEB and hikers' PEB in China's Wuyishan national park while using the TPB mode and NAM in tandem. The following theoretical additions to the existing literature were provided by this investigation: firstly, this study found that both the TPB and NAM demonstrated significant explanatory capacity for tourists' and hikers' PEB in the protected areas, further validating the findings of previous studies on PEB [30, 33, 52, 72, 107–112].

Second, this study has shed more light on hikers' and tourists' PEB. In particular, hikers' ATT, AC, and AR were significantly different from those of tourists. An in-depth review of previous literature revealed that hikers support environmental protection and cherish the surroundings and terrain of their destinations [67, 68]. In other words, hikers' PEB stems primarily from internal demand, as they have a deep connection with nature and want to preserve it. Because hikers value the preserved natural areas in national parks, their PEB is driven by a personal need rather than observance. This conclusion builds on earlier research comparing the pro-environmental intentions of local and non-local tourists, which suggested that the NAM was preferable to the TPB [21].

This study provides management implications for national parks and constructed two integrated models based on TPB and NAM. A more systematic analysis explains the influencing factors on tourists' and hikers' PEB in China. National parks are created to protect the ecological environment, and the protected areas of individual countries form an important part of global ecological protection. In addition, national parks contribute to national nature education, through which visitors learn about the importance of biodiversity and the protection of the ecological environment, thus promoting PEB.

As an increasing number of people participate in hiking, national parks need to accommodate different groups of people and provide different methods and targeted measures to promote their PEB. Tourists might feel less connected to nature, or have no prior experience on pro-environmental practices, which make them less attentive to the environment, resulting in irresponsible behavior towards the environment. National parks should provide education programs on the function of ecological systems, enhance the promotion of environmental protection knowledge, and establish practical do's and don'ts in protecting the environment. In

order to boost tourist enthusiasm, management could also offer specific, targeted incentive programs (e.g., reduce the cost of admission tickets and souvenirs). Additionally, more rubbish bag resupply stations must be put in national parks to encourage tourists to consider themselves as engaging in pro-environmental behavior. Parents should actively engage in pro-environmental conduct and set an example for their children because family activities have a stronger impact on tourists. For the hikers, who they do PEB from their inner psychology, national parks could prompt them spread word-of-mouth, and increase their perceived level of awareness of environmental consequences and responsibility.

## Limitations and future research

There are several limitations to this study that call for additional research, despite the fact that this work offers a fresh perspective on the investigation of hikers' PEB research in China's Wuyishan national park. Firstly, there are a variety of elements that have direct or indirect effects on hikers' PEBs. This study did not consider how all of these variables interacted [98, 113]. Secondly, the results of this study may not be suited for other national parks. In the future, this study could be applied to other different national parks for validation. Thirdly, the study only tested two models; there are many different models that can be used to research PEB, and future research should test the applicability of other models (e.g., the goal-directed behavior model, value-belief-norm theory, self-determination theory, and place attachment theory).

## Conclusions

Negative effects on the environment have gotten more attention as tourism has grown quickly. It is therefore becoming increasingly necessary to identify the factors that influence how tourists and hikers behave in their environment. From the perspective of the behavioral setting, this study examined the TPB and NAM that affected tourists' and hikers' PEB and reached some significant conclusions. After the analysis and model verification of this research, H1-H7 were verified.

First, the findings showed that the PEB of tourists and hikers in China's Wuyishan National Park could be explained by both the TPB and the NAM. Scholars from a variety of disciplines have regularly evaluated the suitability of utilizing TPB or NAM to explain PEB.

Second, attitude, subjective norms, perceived behavioral control, awareness of consequences, ascription of responsibility, and personal norms all influence tourists' and hikers' PEB. ATT, PBC, SN had a positive effect on BI, while AR and AC had a positive impact on PN; AC had a positive influence on AR; PN had a positive influence on BI. These conclusion are consistent with previous research.

Third, in protected areas, such as Wuyishan national park, TPB was more suitable for research on the tourists' PEB, while NAM was more suitable for research on the hikers' PEB. Hikers' ATT, AC, and AR were significantly different from those of the tourists, three factors were play more important than tourist.

## Author Contributions

**Conceptualization:** Qing Zhang.

**Data curation:** Qing Zhang, Qiuyan Lin.

**Formal analysis:** Huazhen Sun.

**Funding acquisition:** Xiasui Peng.

**Investigation:** Huazhen Sun.

**Software:** Xiasui Peng.

**Validation:** Xiasui Peng.

**Writing – original draft:** Qing Zhang.

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
