## [Decision Letter · Decision Letter 0]

30 Mar 2023

PONE-D-23-04451Who take more pro-environmental behavior in national park: Compare the tourist and the hikerPLOS ONE

Dear Dr. Zhang,

Thank you for submitting your manuscript to PLOS ONE. After careful consideration, we feel that it has merit but does not fully meet PLOS ONE’s publication criteria as it currently stands. Therefore, we invite you to submit a revised version of the manuscript that addresses the points raised during the review process.

We look forward to receiving your revised manuscript.

Kind regards,

Bo Pu, Ph.D.

Academic Editor

PLOS ONE

Journal Requirements:

2.  In the ethics statement in the Methods, you have specified that verbal consent was obtained. Please provide additional details regarding how this consent was documented and witnessed, and state whether this was approved by the IRB.

"no"

"no"

Reviewers' comments:

Reviewer's Responses to Questions

**Comments to the Author**

1. Is the manuscript technically sound, and do the data support the conclusions?

Reviewer #1: Partly

Reviewer #2: Yes

2. Has the statistical analysis been performed appropriately and rigorously? 

Reviewer #1: Yes

Reviewer #2: Yes

3. Have the authors made all data underlying the findings in their manuscript fully available?

Reviewer #1: Yes

Reviewer #2: Yes

4. Is the manuscript presented in an intelligible fashion and written in standard English?

Reviewer #1: No

Reviewer #2: Yes

5. Review Comments to the Author

Reviewer #1: Thank you for allowing me to review this manuscript titled “Who take more pro-environmental behavior in national park: Compare the tourist and the hiker”. The paper aims to use Theory of planned behavior and Norm Activation model to explore the intention of tourists’ and hikers' pro-environmental behavior. Overall, the idea is great and provides new knowledge. However, these critical suggestions and corrections are made for the authors to improve the manuscript for possible publication consideration.

Topic

The topic, “Who take more pro-environmental behavior in the national park: Compare the tourist and the hiker” is not scientifically clear. I suggest authors rephrase it and check grammar. However, this is suggested to the reviewer for authors. “Who behaves more pro-environmental at the national park: A comparison of the tourist and the hiker

Abstract.

1. Authors write “The intention of people’s pro-environmental behavior (PEB) directly affects the sustainable development of national park, but less studies do the compare research of tourist and hiker’ This opening statement is not clear to gain readers’ interest. “What national park?

2. This statement is not clear “The results demonstrate that the TPB model and the NAM model were accept of tourists’ and hikers’ pro-environmental behavior in national park, hikers’ attitude, awareness of consequences, ascription of responsibility were significantly different from the tourist”

3. Authors should provide a sentence of policy implication in the last part of abstract

4. Authors should define “ERB” in its first appearance

5. Abstract has a serious language problem

Introduction

1. “…Therefore, given the unique characteristics of mountainous areas, increasing PEB of tourists has emerged as a crucial problem that requires immediate resolution” Authors must provide the unique characteristics of the mountainous areas they talk about.

2. This statement “China is an emerging and vast tourism destination, more than 67% of China's land area is made up of mountainous regions [13], as scenic mountain area attracts large numbers of tourists and hikers, the problem of environmental damage in these areas is getting worse which is also facing various environmental issues” is too long and must be broken down and made clearer.

3. People believed to walk in mountainous areas with different altitudes, which can bring many benefits to participants. What happened today? They don’t believe anymore? Please make statement clear.

4. “As of July 2022, China has a total of 5 National Parks: Sanjiangyuan, Panda, Northeast Tiger and Leopard, Hainan Tropical Rainforest, Wuyishan National Parks.” Please cite a source.

5. What do authors mean by “The management of national parks must be sustainable?

6. Authors claim to do comparative research of a tourist and hiker but only talks about the hikers. Are hikers not tourists? What is the difference between a tourist and a hiker?

7. Authors must highlight clearly the theoretical and practical contribution of the study in the introduction

8. Authors must explain in the introduction what actions they consider pro-environmental at national parks

Literature review

1. Section 2 should be Literature review and hypotheses development not “Materials and Methods”

2. Literature review is well written

3. Provide a conceptual framework diagram showing the various association you intend to find to increase readability

Materials and Methods

1. Authors can change Section 3 to “Data, materials, and Methods

2. Make this statement clearer ‘Wuyishan National Park is a world natural and cultural heritage recognized by United Nations Educational, Scientific and Cultural Organization (UNESCO), there are only four places in China”

3. In 2022, more than one million tourists “come” (came) to Wuyishan National Park for visit.

4. Section 3.2, Based on the language problems in the manuscript, I believe there were misunderstanding issues in the questionnaire designs. Having three professors translate questionnaire. Did the professors check the language for this manuscript as well?

5. Data collection method is well written

6. Was data collected for only Chinese citizens or foreigners included? Please explain

Results.

1. Provide table for demographics of both respondent category

2. Measurement model should be in Section 3 and only the results p[resented in section 4

3. Table 2: As suspected there are grammatical issues with the questionnaire items. Example: “I think do PEB in national park is wise” is not clear and can affect respondents response”

4. Hypotheses test results has not been clearly reported.

Discussion, Conclusion and limitations

1. Conclusion should be the last section.

2. Avoid citations in policy implication and conclusions

3. Policy implication is scanty done and need improvement. It should focus on what must be done in real life based on the findings from this study

4. There are several limitations that call for additional research, despite the fact that this work offers a fresh viewpoint on the investigation of hikers' electroreceptor bands… What does this mean?

General comments

1. The manuscript has a serious language problem. There is grammatical error in almost every line which makes it difficult to understand what authors are putting across. Authors need to use a professional language editing tool to significantly improve it

2. Provide line numbers for easy identification and referencing

3. The full meaning of NAM is Norm activation Model, so authors cannot write NAM model which makes it a repetition of “model”

4. For this paper to be considered, it requires a serious improvement especially the language

Reviewer #2: 1.The introduction section lacks much information. So I suggest rewriting the introduction section. The introduction may include five sub-sections. The first part proposes the research topic and introduces the importance of the research topic. The second part carries on the literature review of research topic and points out the research gaps. The third part puts forward some research questions aiming at the research gaps. The fourth part points out the contribution of this research after solving the research questions. The fifth part points out the layout of the article. However, it seems that the introduction did not clearly present the research gap. Meanwhile, why did you select these variables?

2.In addition, The figure of conceptual model is necessary.

3.It have some small details need to be modified, on the page 6, (2) Subjective norms, looks like it should be Subjective norm.

4.In the whole paper, the decimal point of the data is consistent at the end. It is recommended to keep three digits after the decimal point throughout the paper, as shown in Table 1, GFI of the tourist in TPB model is 0.98, it should be change into 0.980. Research methodoly is right.

6. PLOS authors have the option to publish the peer review history of their article (what does this mean?). If published, this will include your full peer review and any attached files.

Reviewer #1: No

Reviewer #2: No

<quillbot-extension-portal></quillbot-extension-portal>

---

## [Author Response · Author response to Decision Letter 0]

28 Apr 2023

Response to Reviewer 1 Comments

We would first like to thank you for all the constructive and critical comments on our paper. When revising this paper, we would like to assure you that we have seriously considered all comments and diligently addressed all concerns as thoroughly as possible. Regarding your comments, we have addressed them in our revision as follows.

Topic

Point 1: The topic, “Who take more pro-environmental behavior in the national park: Compare the tourist and the hiker” is not scientifically clear. I suggest authors rephrase it and check grammar. However, this is suggested to the reviewer for authors. “Who behaves more pro-environmental at the national park: A comparison of the tourist and the hiker

Response 1: The study followed with your suggestion, we changed the topic to “Who behaves more pro-environmental in the national parks: A comparison of the tourist and the hiker” 

Abstract. 

Point 1: Authors write “The intention of people’s pro-environmental behavior (PEB) directly affects the sustainable development of national park, but less studies do the compare research of tourist and hiker’ This opening statement is not clear to gain readers’ interest. “What national park?

Response 1: we add more detail. “The intention of people’s pro-environmental behavior (PEB) directly affects the sustainable development of protected areas, especially for the national park.” (line 8-9)

Point 2: This statement is not clear “The results demonstrate that the TPB model and the NAM model were accept of tourists’ and hikers’ pro-environmental behavior in national park, hikers’ attitude, awareness of consequences, ascription of responsibility were significantly different from the tourist”

Response 2: we rewrite it. “The results demonstrate that the TPB model and the NAM model were accept for tourists’ and hikers’ pro-environmental behavior in national park. However, for the specific influencing factors, hikers’ attitude, awareness of consequences, ascription of responsibility were significantly different from the tourist.” (line 14-17)

Point 3: Authors should provide a sentence of policy implication in the last part of abstract

Response 3: we rewrite it and add a sentence “This study sheds light on how to better comprehend and advocate for PEB in national parks and proposes different management approaches to improve the PEB of tourists and hikers.” (line 17-20)

Point 4: Authors should define “ERB” in its first appearance

Response 4: It is PEB, we wrote the wrong words. 

Point 5: Abstract has a serious language problem

Response 5: The study ask the company “Cambridge Proofreading” to chick the English written for academic journal, and we also do the chick again.

Introduction

Point 1: “…Therefore, given the unique characteristics of mountainous areas, increasing PEB of tourists has emerged as a crucial problem that requires immediate resolution” Authors must provide the unique characteristics of the mountainous areas they talk about.

Response 1: we add the sentence. “The mountain is particularly precarious with more fragile, ecological environment, and it is more difficult to collect and transfer litter [2, 3, 11].” (line 31-33)

Point 2: This statement “China is an emerging and vast tourism destination, more than 67% of China's land area is made up of mountainous regions [13], as scenic mountain area attracts large numbers of tourists and hikers, the problem of environmental damage in these areas is getting worse which is also facing various environmental issues” is too long and must be broken down and made clearer.

Response 2: we broken down the long sentence, and make it clearer. In line 30, “China is a vast, emerging tourism destination in which more than 67% of land area is comprised of mountainous regions [10].” (line 30-31) 

“This has resulted in worsening conditions and environmental challenges, as picturesque mountain areas draw significant numbers of hikers [21, 22].” (line 42-44)

Point 3: People believed to walk in mountainous areas with different altitudes, which can bring many benefits to participants. What happened today? They don’t believe anymore? Please make statement clear.

Response 3: sorry, we used the wrong verb tenses, they believed it in the past time, and today they still believe. “People believe to walk in mountainous areas with different altitudes, which can bring many benefits to participants [17]” (line 39-41)

Point 4: “As of July 2022, China has a total of 5 National Parks: Sanjiangyuan, Panda, Northeast Tiger and Leopard, Hainan Tropical Rainforest, Wuyishan National Parks.” Please cite a source.

Response 4: we add the cite. “As of July 2022, China had a total of 5 National Parks: Sanjiangyuan, Panda, Northeast Tiger and Leopard, Hainan Tropical Rainforest, and Wuyishan [12]”(line 33-35)

Point 5: What do authors mean by “The management of national parks must be sustainable?

Response 5: we rewrite the sentence. “The national parks need to be profitable and environmentally friendly.”(line 35-36)

Point 6: Authors claim to do comparative research of a tourist and hiker but only talks about the hikers. Are hikers not tourists? What is the difference between a tourist and a hiker?

Response 6: we add the sentences. “Traditional visitors have been use as research subjects in the majority of previous studies, but hikers’ present unique characteristics; for example, hikers are typically better educated and more environmentally conscious [66] , supporting nature conservation and valuing natural beauty and landscapes [67, 68]”(line 57-60)

Point 7: Authors must highlight clearly the theoretical and practical contribution of the study in the introduction.

Response 7: we add the sentences. “This study made an in-depth comparison of the PEB of tourists and hikers, using TPB and NAM for detection, and comparative analysis of the factors in the models. Finally, we propose different management approaches for tourists and hikers to improve PEB.”(line 82-85)

Point 8: Authors must explain in the introduction what actions they consider pro-environmental at national parks

Response 8: we add the sentences. “This study used Wuyishan national park as a case study, taking token littering, stepping on grass, picking of flowers, and breaking of branches, plants and trees as irresponsible environmental behavior [22]”(line 69-72)

Literature review

Point 1: Section 2 should be Literature review and hypotheses development not “Materials and Methods”

Response 1: we change the section 2 to the “A Literature review and hypotheses development”(line 86)

Point 2: Literature review is well written

Response 2: Thank you for your comment.

Point 3: Provide a conceptual framework diagram showing the various association you intend to find to increase readability

Response 3: we add the two conceptual framework diagram. Fig. 1. The theory of planned behavior framework (line 168-173); Fig. 2. The norm activation theory framework (line 175-181)

Materials and Methods

Point 1: Authors can change Section 3 to “Data, materials, and Methods

Response 1: we change Section 3 to “Data, materials, and Methods” (line 182)

Point 2: Make this statement clearer ‘Wuyishan National Park is a world natural and cultural heritage recognized by United Nations Educational, Scientific and Cultural Organization (UNESCO), there are only four places in China”

Response 2: we rewrite the sentence. “Wuyishan National Park is one of the four world natural and cultural heritage sites recognized by the United Nations Educational, Scientific and Cultural Organization (UNESCO) in China.”(line 184-186)

Point 3: In 2022, more than one million tourists “come” (came) to Wuyishan National Park for visit.

Response 3: we rewrite the sentence. “In 2022, more than one million tourists came to visite Wuyishan National Park.”(line 188-189)

Point 4: Section 3.2, Based on the language problems in the manuscript, I believe there were misunderstanding issues in the questionnaire designs. Having three professors translate questionnaire. Did the professors check the language for this manuscript as well?

Response 4: The professors' translation and grammar proofreading are only for the questionnaire, and the expert does not help check the language for this manuscript. The study ask the company “Cambridge Proofreading” to chick the English written for academic journal, and we also do the chick again.

Point 5: Data collection method is well written

Response 5: Thank you for your comment.

Point 6: Was data collected for only Chinese citizens or foreigners included? Please explain

Response 6: we rewrite the sentence. “The questionnaire was distributed to Chinese tourists and Chinese hikers by way of convenience sampling in the Wuyishan National Park”(line 229-230)

Results.

Point 1: Provide table for demographics of both respondent category

Response 1: we add the table 1 Demographic data(line 269)

Point 2: Measurement model should be in Section 3 and only the results presented in section 4

Response 2: We moved the Measurement model testing to the Section 3,and in Section 4 just presented the results.

Point 3: Table 2: As suspected there are grammatical issues with the questionnaire items. Example: “I think do PEB in national park is wise” is not clear and can affect respondents response”

Response 3: we chick every questionnaire items for the English written for academic journal and make it more clear.

Point 4: Hypotheses test results has not been clearly reported.

Response 4: we add the sentence. “ATT, PBC, SN have a positive effect on BI (tourist:β= 0.197, p＜0.001; β=0.243, p＜0.001; β=0.207, p＜0.05; hiker: β= 0.422, p＜0.001; β=0.171, p＜0.001; β=0.294, p＜0.05), H1, H2, and H3 were all supported; AR and AC have a positive impact on PN (tourist:β=0.195, p<0.001; β=0.271, p<0.001; hiker:β=0.558, p<0.001; β=0.318, p<0.001), H4 and H5 were all supported; AC has a positive influence on AR (tourist:β=0.315, p<0.001; hiker:β=0.667, p<0.001), H6 was supported; PN has a positive influence on BI (tourist:β=0.443, p<0.001; hiker:β=0.421, p<0.001), H7 was supported.”(line 292-300)

Discussion, Conclusion and limitations

Point 1: Conclusion should be the last section

Response 1: We moved the conclusion to the last section

Point 2: Avoid citations in policy implication and conclusions

Response 2: We removed reference in policy implication and conclusions

Point 3: Policy implication is scanty done and need improvement. It should focus on what must be done in real life based on the findings from this study

Response 3: we add the sentence. “For the tourists, they might feel less connected to the nature, or have no prior experience on pro-environmental practices, which make them less attentive to the environment, resulting in irresponsible behavior towards the environment. National parks should provide education programs on the function of ecological system, enhance the promotion of environmental protection knowledge, and establish the practical dos and don’ts in protecting the environment. In order to boost tourist enthusiasm, management could also offer specific targeted incentive programs (e.g., reduce the cost of admission tickets and souvenirs). Additionally, more rubbish bag resupply stations must to be put in national park to encourage tourists to consider themselves as having pro-environmental behavior. Parents should actively engage in pro-environmental conduct and set an example for their children because family activities have a stronger impact on tourists. For the hikers, they do the PEB from their, national park need could encourage spread word-of-mouth, and increasing their perceived level of wareness of environmental consequences and responsibility.”(line 395-401)

Point 4: There are several limitations that call for additional research, despite the fact that this work offers a fresh viewpoint on the investigation of hikers' electroreceptor bands… What does this mean?

Response 4: we rewrite the sentence.”There are several limitations to this study that call for additional research, despite the fact that this work offers a fresh perspective on the investigation of hikers' PEB research in China's Wuyishan national park.” (line 405-407)

General comments

Point 1: The manuscript has a serious language problem. There is grammatical error in almost every line which makes it difficult to understand what authors are putting across. Authors need to use a professional language editing tool to significantly improve it

Response 1: The study ask the company “Cambridge Proofreading” to chick the English written for academic journal, and we also do the chick again.

Point 2: Provide line numbers for easy identification and referencing

Response 2: We add the line numbers for this study.

Point 3: The full meaning of NAM is Norm activation Model, so authors cannot write NAM model which makes it a repetition of “model”

Response 3: We do the chick, and change NAM model to NAM.

Point 4: For this paper to be considered, it requires a serious improvement especially the language.

Response 4: The study ask the company “Cambridge Proofreading” to chick the English written for academic journal, and we also do the chick again.

Response to Reviewer 2 Comments

We would first like to thank you for all the constructive and critical comments on our paper. When revising this paper, we would like to assure you that we have seriously considered all comments and diligently addressed all concerns as thoroughly as possible. Regarding your comments, we have addressed them in our revision as follows.

Point 1: The introduction section lacks much information. So I suggest rewriting the introduction section. The introduction may include five sub-sections. The first part proposes the research topic and introduces the importance of the research topic. The second part carries on the literature review of research topic and points out the research gaps. The third part puts forward some research questions aiming at the research gaps. The fourth part points out the contribution of this research after solving the research questions. The fifth part points out the layout of the article. However, it seems that the introduction did not clearly present the research gap. Meanwhile, why did you select these variables?

Response 1: we follow with your suggestion, rewrite the introduction. (line 26-85). We made the research gap more clear .

Point 2: In addition, The figure of conceptual model is necessary.

Response 2: we add the two conceptual framework diagram. Fig. 1. The theory of planned behavior framework (line 168-173); Fig. 2. The norm activation theory framework (line 175-181)

Point 3: It have some small details need to be modified, on the page 6, (2) Subjective norms, looks like it should be Subjective norm.

Response 3: Thank you for your comment, we do the chick the small details again. 

Point 4: In the whole paper, the decimal point of the data is consistent at the end. It is recommended to keep three digits after the decimal point throughout the paper, as shown in Table 1, GFI of the tourist in TPB model is 0.98, it should be change into 0.980. Research methodoly is right.

Response 4: Thank you for your comment, we do the chick again, keep three digits after the decimal point.

---

## [Decision Letter · Decision Letter 1]

12 May 2023

PONE-D-23-04451R1Who behaves more pro-environmental in the national parks: A comparison of the tourist and the hikerPLOS ONE

Dear Dr. Zhang,

Thank you for submitting your manuscript to PLOS ONE. After careful consideration, we feel that it has merit but does not fully meet PLOS ONE’s publication criteria as it currently stands. Therefore, we invite you to submit a revised version of the manuscript that addresses the points raised during the review process.

We look forward to receiving your revised manuscript.

Kind regards,

Bo Pu, Ph.D.

Academic Editor

PLOS ONE

Journal Requirements:

Reviewers' comments:

Reviewer's Responses to Questions

**Comments to the Author**

1. If the authors have adequately addressed your comments raised in a previous round of review and you feel that this manuscript is now acceptable for publication, you may indicate that here to bypass the “Comments to the Author” section, enter your conflict of interest statement in the “Confidential to Editor” section, and submit your "Accept" recommendation.

Reviewer #1: All comments have been addressed

Reviewer #2: (No Response)

2. Is the manuscript technically sound, and do the data support the conclusions?

Reviewer #1: Yes

Reviewer #2: Yes

3. Has the statistical analysis been performed appropriately and rigorously? 

Reviewer #1: Yes

Reviewer #2: Yes

4. Have the authors made all data underlying the findings in their manuscript fully available?

Reviewer #1: Yes

Reviewer #2: Yes

5. Is the manuscript presented in an intelligible fashion and written in standard English?

Reviewer #1: Yes

Reviewer #2: Yes

6. Review Comments to the Author

Reviewer #1: Find attached my comments. Authors have improved the paper but another round of proofreading will be great

Reviewer #2: Thank you for your effort for revising this study. The authors responsed and revised the question very well. However, I suggest the authors combine the figure 1 and figure 2 as a whole conceptual model.

7. PLOS authors have the option to publish the peer review history of their article (what does this mean?). If published, this will include your full peer review and any attached files.

Reviewer #1: No

Reviewer #2: No

While revising your submission, please upload your figure files to the Preflight Analysis and Conversion Engine (PACE) digital diagnostic tool, https://pacev2.apexcovantage.com/. PACE helps ensure that figures meet PLOS requirements. To use PACE, you must first register as a user. Registration is free. Then, login and navigate to the UPLOAD tab, where you will find detailed instructions on how to use the tool. If you encounter any issues or have any questions when using PACE, please email PLOS at figures@plos.org. Please note that Supporting Information files do not need this step.<quillbot-extension-portal></quillbot-extension-portal>

---

## [Author Response · Author response to Decision Letter 1]

14 May 2023

Response to Reviewer 1 Comments

We would first like to thank you for all the constructive and critical comments on our paper. When revising this paper, we would like to assure you that we have seriously considered all comments and diligently addressed all concerns as thoroughly as possible. Regarding your comments, we have addressed them in our revision as follows.

Point 1: Check papers published in the journal for proper outline and structure of manuscript

Response 1: Thank you for your comment, we download the papers published in the Plos one for proper outline and structure of manuscript.

Point 2: Line 39-41 “People believe to walk in mountainous areas with different altitudes, which can bring many benefits to” is still not clear. This is a suggestion. 

“People believe walking in mountainous areas with different altitudes can bring many benefits to them” 

OR

“It is believed that walking in mountainous areas with different altitudes, can bring many benefits to the participants.”

Response 2: we used “It is believed that walking in mountainous areas with different altitudes, can bring many benefits to the participants.” Replace the “People believe to walk in mountainous areas with different altitudes, which can bring many benefits to”

Point 3: Errror: Traditional visitors have been used as research subjects in the majority of previous studies, but hikers’ present unique characteristics….

Response 3: we rewrite it. “Traditional visitors have been used as research subjects in the majority of previous studies, but hikers’ present unique characteristic” 

Point 4: Authors must number sections and subsections

1. Introduction

2. Literature and hypotheses

3. Methods

4. Etc

Response 4: we download the paper published in the Plos one, and found that it do not need to add the number sections and subsections.

Response to Reviewer 2 Comments

We would first like to thank you for all the constructive and critical comments on our paper. When revising this paper, we would like to assure you that we have seriously considered all comments and diligently addressed all concerns as thoroughly as possible. Regarding your comments, we have addressed them in our revision as follows.

Point 1: The authors responsed and revised the question very well. However, I suggest the authors combine the figure 1 and figure 2 as a whole conceptual model.

Response 1: we combine the figure 1 and figure 2 as a whole conceptual model. See the figure 1 Conceptual model.

---

## [Decision Letter · Decision Letter 2]

2 Jun 2023

Who behaves more pro-environmental in the national parks: A comparison of the tourist and the hiker

PONE-D-23-04451R2

Dear Dr. Zhang,

We’re pleased to inform you that your manuscript has been judged scientifically suitable for publication and will be formally accepted for publication once it meets all outstanding technical requirements.

Kind regards,

Bo Pu, Ph.D.

Academic Editor

PLOS ONE

Additional Editor Comments (optional):

this manuscript will be accepted for publication in PLOS ONE.

---

## [Editor Report · Acceptance letter]

15 Jun 2023

PONE-D-23-04451R2 

Who behaves more pro-environmental in the national parks: A comparison of the tourist and the hiker 

Dear Dr. Zhang:

I'm pleased to inform you that your manuscript has been deemed suitable for publication in PLOS ONE. Congratulations! Your manuscript is now with our production department. 

Kind regards, 

on behalf of

Dr. Bo Pu 

Academic Editor

PLOS ONE